# Superconductor-insulator transition in space charge doped one unit cell $Bi_{2.1}Sr_{1.9}CaCu_2O_{8+x}$

Fang Wang[1], Johan Biscaras[1 ✉], Andreas Erb[2] & Abhay Shukla [1 ✉]

The superconductor-insulator transition in two dimensions is a prototype continuous quantum phase transition at absolute zero, driven by a parameter other than temperature. Here we reveal this transition in one unit-cell $Bi_{2.1}Sr_{1.9}CaCu_2O_{8+x}$ by space charge doping, a field effect electrostatic doping technique. We determine the related critical parameters and develop a reliable way to estimate doping in the nonsuperconducting region, a crucial and central problem in these materials. Finite-size scaling analysis yields a critical doping of 0.057 holes/Cu, a critical resistance of ~6.85 k$\Omega$ and a scaling exponent product $\nu z$ ~ 1.57. These results, together with earlier work in other materials, provide a coherent picture of the superconductor-insulator transition and its bosonic nature in the underdoped regime of emerging superconductivity in high critical temperature superconductors.

[1] Sorbonne Université, CNRS UMR7590, MNHN, Institut de Minéralogie, de Physique des Matériaux et de Cosmochimie, IMPMC, Paris, France. [2] Walther Meissner Institut fur Tieftemperaturforschung, Bayerische Akademie der Wissenschaften, Garching, Germany. ✉email: johan.biscaras@sorbonne-universite.fr; abhay.shukla@sorbonne-universite.fr

A superconductor-insulator transition (SIT) in two dimensions is a continuous quantum phase transition (QPT) at absolute zero temperature[1,2] driven by external parameters like disorder, magnetic field, or carrier concentration[3–11].

Such transitions have been induced in a variety of 2D superconductors by tuning different external parameters and studied with a finite-size scaling analysis. There is however not much uniformity in the findings as both the superconducting systems and the tuning parameters are diverse. Magnetic field is a common, easily tunable and accessible external parameter. Magnetic field driven SIT studies have reported a large variety of critical exponents and resistance at criticality: from $\nu z \sim 0.67$ and $R_c = 1.3\,k\Omega$ in NbSi thin films[12] to $\nu z \sim 2.3$ and $R_c = 6\,k\Omega$ in Indium oxide thin films[13]. Furthermore, in quenched condensed bismuth thin films the magnetic field driven SIT was shown to have $\nu z \sim 0.7$ and $R_c = 8\,k\Omega$, while the thickness driven SIT displayed $\nu z \sim 1.2$ around the same critical resistance, indicating a fundamental difference in their nature[3]. It is therefore difficult to come to a general conclusion about the SIT in different systems and induced by different tuning parameters. The high critical magnetic fields of most high critical temperature superconductors renders the study of the magnetic field driven SIT difficult with the exception of the electron doped cuprate NdCeCuO[14]. However, improvements in extreme electrostatic doping techniques have provided access to the carrier density driven SIT in $La_{2-x}Sr_xCuO_4$ (LSCO) and $YBa_2Cu_3O_{7-x}$ (YBCO).

Tuning carrier density is an effective way to realize this transition by chemical or electrostatic doping methods. Chemical doping, the usual and only possible method in bulk samples, has been widely used in high critical temperature superconductors, both in bulk and thin films[11,15]. However it is sample dependent and can lead to disorder and structure change. Electrostatic doping is the method of choice to obtain a continuous and defect free change in doping in the same ultrathin sample in a field effect transistor device[7,8,16]. Both the correlation length $\epsilon$ and correlation time $\tau$ corresponding to a phase transition are dependent on the variation of an external parameter $x$ with respect to a critical value $x_c$. In our case this is the variation of the carrier concentration $p$ with respect to the critical doping at the phase transition $p_c$ with, $\epsilon \propto |p - p_c|^{-\nu}$ and $\tau \propto \epsilon^z$ where $\nu$ is the correlation length exponent and $z$ is the dynamical-scaling exponent. The variation of physical quantities like sheet resistance across the phase transition in a transport measurement can be expressed in terms of these asymptotic forms and a single "scaling" formula dependent on $|p - p_c|^{-\nu/z}$. If the effects of the QPT persist at experimentally accessible, small, but non-zero temperatures, it can be characterized by its universality class given by the numerical value of $\nu z$, the product of the finite-scaling exponents. This value, along with the critical values of the driving parameter (doping) and the measured physical quantity (sheet resistance), consitutes the fundamental information that can be gleaned from a QPT in reduced dimensions and has been used in the past for studying high $T_c$ superconductors[7,8,11].

In this work we establish and investigate the nature of the SIT as a function of doping in two dimensional $Bi_{2.1}Sr_{1.9}CaCu_2O_{8+x}$ (BSCCO) and deduce the scaling parameters associated with this QPT. This measurement has been rarely accomplished, always with considerable experimental process. The QPT has been shown to exist in two of the principal families of high $T_c$ compounds, LSCO[7] and YBCO[8] using ultrathin samples and electrostatic doping. In BSCCO a recent result in single layer (i.e., half unit cell) samples[11] is remarkable but uses chemical doping with ozone. Here we accomplish electrostatic doping with our space charge doping technique on a one unit cell (1 u.c.) BSCCO device and indeed observe a QPT. Earlier theoretical work[17] predicted the existence of a material-independent quantum critical resistance $R_Q = h/(2e)^2 = 6.45\,k\Omega$. $\square^{-1}$ for the insulator-superconductor transition. Experimental results from the above works show some scatter, with near universal critical resistance values in LSCO (6.4 k$\Omega$)[7] and YBCO (6 k$\Omega$)[8] but a variation of 2.8 to 10.2 k$\Omega$ in BSCCO[11]. The critical doping associated with this crossover varies significantly, from $p_c = 0.05$–$0.06$ holes/Cu[7,8], the value expected from the generic phase diagram, to nearly 0.02 holes/Cu[11]. Finally the product of critical exponents $\nu z$ also varies, with 1.5 in LSCO[7], 2.2 in YBCO[18], and between 1.5 and 2.4 in BSCCO[11]. Universality of the QPT should imply $R_c \sim 6.45\,k\Omega$, $p_c \sim 0.05$–$0.06$, and a similar value for $\nu z$ in different materials. Thus establishing this benchmark in BSCCO is necessary and the measurement should avoid pitfalls from sample dependent imperfections which tend to overshadow material parameters.

A crucial aspect of this program is determining the doping since absolute determination of doping in high $T_c$ compounds, whether bulk or few layer samples, remains elusive. The Hall coefficient is notoriously variable with temperature. In LSCO and YBCO[7,8] the inverse sheet resistance at a fixed temperature well above the $T_c$ is proposed as a measure of the doping $p$, presupposing simple Drude-like behavior. An empirical relation, linking the dome shaped dependence of the critical temperature on doping[9,19,20], has also been used. Can this be extended to the strongly undoped nonsuperconducting region? We show here that it can and indeed gives a reliable, though not absolute, determination of doping which could profitably be used in future work.

## Results

Bulk crystals of BSCCO, ($T_c = 89$ K), were used to fabricate large area, ultrathin 1 u.c. BSCCO samples of $\sim100\,\mu m$ lateral size on 0.5 mm thick soda-lime glass substrate by the anodic bonding method[9,21,22]. The 1 u.c. sample oxygen stoichiometry was initially reduced to the region of the beginning of the superconducting dome by annealing the sample in air at 350 °C. The variation of the doping level was then achieved through space charge doping[23,24] and monitored by controlling the $R_s$ through the change of gate voltage at high temperature. Transport measurements were performed from 330 K to low temperatures. Details are provided in the methods section. The device used in this work is labeled F. For comparison similar devices from earlier work labeled C, D[9], and E[10] are also referred to below.

Figure 1b shows the SIT by measuring the $R_S(T)$ curves for varying doping in device F. The thick olive green curve indicates that the starting doping level $p$ after annealing is indeed near the beginning of the superconducting dome. The corresponding critical temperature $T_c$ (defined in this work as the temperature below which $R_S$ vanishes) is near 10 K. The doping is then tuned step by step as shown in Supplementary Fig. 1. The immediate challenge, elusive even in bulk high $T_c$ superconductors, is to determine the doping level. In a simple single band model the Hall coefficient $R_H$ and the elementary charge $q$ directly give the charge carrier concentration (or doping level) $p = 1/qR_H$. High $T_c$ superconductors can certainly not be classified as such and indeed $R_H$ is anomalously temperature dependent. Another possible estimation is through the classical Drude model where conductivity is given by the product of the elementary charge, the mobility, and the carrier concentration. Based on this, attempts have been made to determine the carrier concentration using $p = S/R_S(T_f)$ where $S$ is an empirically determined constant and $R_S(T_f)$ the sheet resistance at a fixed temperature well above $T_c$[7,8]. The constant $S$ is determined by the maximum of the superconducting dome which is nominally set at 0.16 holes/Cu. In the inset of Fig. 2a we show the relation between the Hall ($1/qR_H$) and Drude

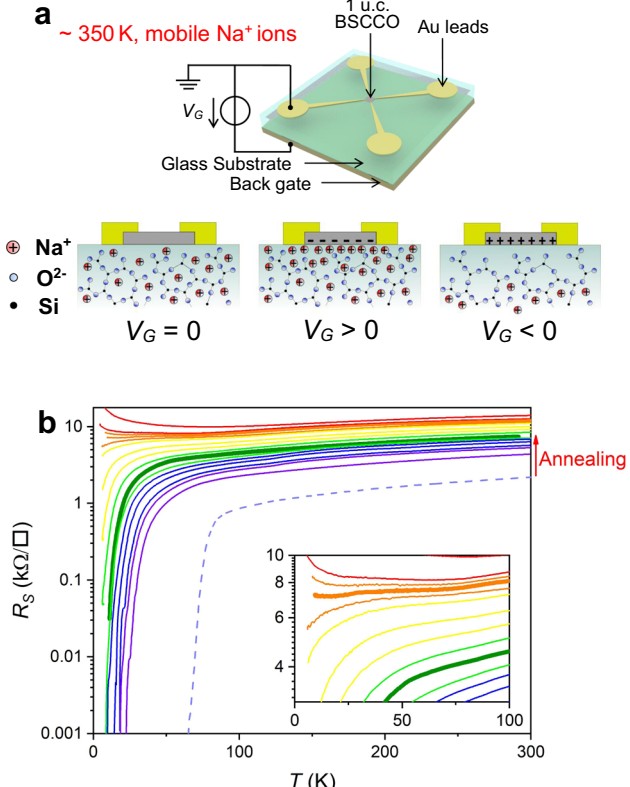

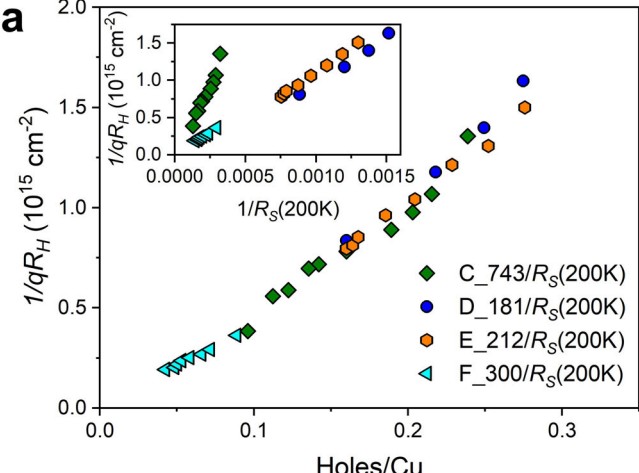

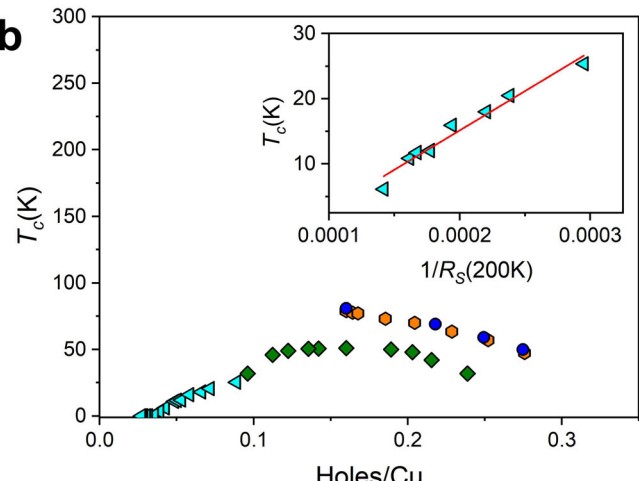

**Fig. 1 Superconductor-insulator transition driven by space charge doping.** **a** Schematic view of the device and of the space charge doping method[23]. **b** Temperature dependence of $R_S$ curves induced by space charge doping for device F. The thick olive green $R_S(T)$ curve is the initial sheet resistance of the sample after annealing to reduce doping whereas the dashed violet curve is representative of the optimal doping level of 1 u.c. BSCCO after fabrication by anodic bonding. The inset shows a blow-up and the thick orange "horizontal" curve approximately corresponding to the critical doping.

**Fig. 2 Hall coefficient and inverse sheet resistance for estimating doping.** **a** Dependence of $1/qR_H$ on $p = S/R_S(T_f)$ for device F of this work and devices C, D, and E of our earlier works. Inset: Dependence of $1/qR_H$ on $1/R_S(200\,K)$ showing linear behavior. **b** Superconducting dome as a function of $T_c$ and of doping $p$ as calculated above for devices C, D, E, and F. Inset: Critical temperature $T_c$ as a function of $1/R_S(200\,K)$ showing a linear dependence in the neighborhood of zero $T_c$.

$(1/R_S(T_f))$ estimates of doping for device F and devices C, D, and E used in our earlier studies. Two deductions can be made. The relation is linear, implying simple proportionality. Device C, known to have high disorder and low mobility has a markedly different slope, showing the importance of mobility. If the empirical constant $S$ is adjusted for each sample as in Fig. 2a, we can hope to determine $p = S/R_S(T_f)$ as shown on the x axis while maintaining the near linear relation with inverse Hall coefficient. How good is this estimate for BSCCO? In Fig. 2b the critical temperatures of these devices are shown as a function of the above estimation of doped charge. The generic dome shape is flattened and skewed, with superconductivity starting at a low doping of 0.028, and stretching above $p = 0.3$, well beyond the limits of the generic phase diagram. Though this result is coherent with a similar recent determination[11] we conclude that this estimate of doped charge is unsatisfactory.

Another often used approach seeks to estimate doped holes ($p$) per Cu atom with an empirical $T_c(p)$ relation[19] for the region of the superconducting dome.

$$T_c(p)/T_c(p_{opt}) = 1 - Z(p - p_{opt})^2 \qquad (1)$$

where $T_c(p_{opt})$ is the maximum critical temperature measured corresponding to the optimal doping level and $Z$ is a scale factor empirically determined to be 82.6[20,25,26]. In this approach, in accord with the generic phase diagram, superconductivity exists in the region $p \sim 0.05$ to $p \sim 0.27$ holes/Cu. The dome shape

implied by this relation is experimentally verified (notwithstanding local deviations for example in YBCO around $p = 0.12$ doping), in particular for 1 u.c. BSCCO[9]. The problem in our case is that doping has to be determined for nonsuperconducting as well as superconducting regions. To overcome this hurdle we seek inspiration from the simpler approaches discussed above. Firstly we remark (inset Fig. 2b) that when the $T_c$ in the superconducting region is plotted against the inverse sheet resistance at 200 K, a simple linear relationship is found. Extrapolating this relation to the nearby non superconducting region, we replace $T_c$ by $S/R_S(200\,K)$ in Eq. (1) where $S$ is the value of the slope of this linear dependence. We thus have a coherent and continuous estimation of $p$ across the SIT which is compatible with the generic high $T_c$ phase diagram. This method should be applicable to other high $T_c$ materials. In Fig. 3 we show the part of the phase diagram around the beginning of the superconducting dome, relevant to device F. Superconductivity develops according to this estimation at a critical doping of $p \sim 0.057$. In the inset of Fig. 3 we show the superconducting phase diagrams for devices C, D, E, and F, which are compatible with the generic phase diagram by

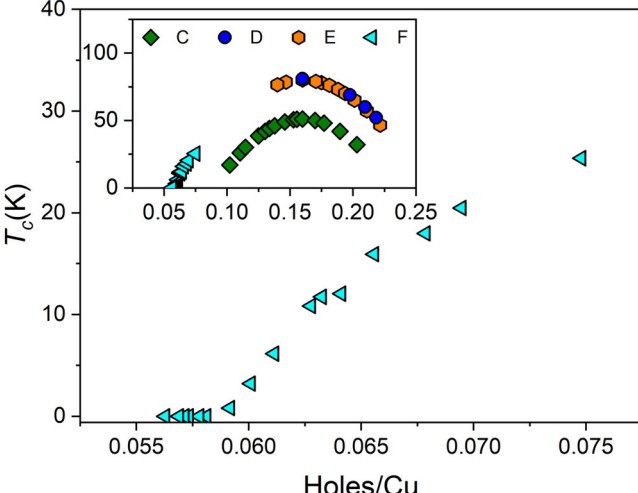

**Fig. 3 Superconducting phase diagram.** Superconducting dome as a function of critical temperature and doping. Inset: Superconducting phase diagram for four devices C, D, E, and F.

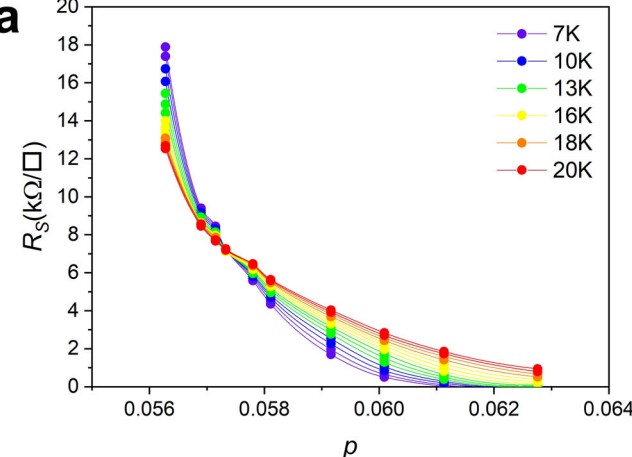

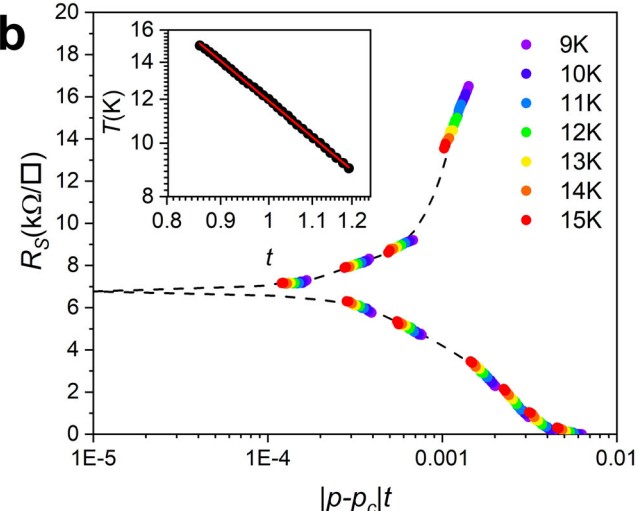

**Fig. 4 Finite-size scaling of QPT. a** Doping dependence of isothermal sheet resistance from 7 to 20 K. The dots are extracted from $R_S(T)$ data and the lines are splined guides for the eye. **b** Universal finite-size scaling function at the SIT with $t = T^{-1/\nu z}$. Inset: The linear relation between $T$ and $t$ and between 9 and 15 K when the data is plotted on a log-log scale. From the slope we obtain the critical exponent product $\nu z = 1.57 \pm 0.10$.

construction and exhibit a smooth variation with an optimum doping of 0.16 holes[9,10]. Remarkably, in a recent study[27] an estimation of doped holes in bulk BSCCO using Fermi surface volume measured by photoemission measurements finds that the dome in BSCCO is smooth, with a range corresponding to our observations and in agreement with the generic high $T_c$ shape.

With these important observations we can set about investigating the existence of possible scaling relations corresponding to a QPT in our 1 u.c. BSCCO device at the SIT. This implies that all sheet resistance curves of Fig. 1b should collapse onto a single finite-size scaling function $R_S = R_c f(|x - x_c|T^{-1/\nu z})$, where $R_c$ is the critical resistance at the limit $x \to x_c$ and $T \to 0$. $f$ is the universal scaling function and $x$ is the tuning parameter, in our case the doping level $p$. The correlation length exponent $\nu$ and the dynamic critical exponent $z$, together with the critical resistance $R_c$ encode the nature of this transition[7,8]. Figure 4 shows the results of finite-size scaling analysis of the SIT in device F. The first indication of critical behavior can be found by plotting the isothermal sheet resistance of the data in Fig. 1b, not as a function of temperature, but as a function of the doping $p$. This is done in Fig. 4a and shows that between 9 and 20 K, all curves intersect at a single point which corresponds to the critical doping $p_c \sim 0.057$ where superconductivity sets in. The corresponding critical sheet resistance is ~6.85 k$\Omega$. $\square^{-1}$. In Fig. 4b we investigate the existence of a single finite-size scaling function $f$ for data between 9 and 15 K by using as the abscissa $|p - p_c|t$. Isothermal sheet resistance data as a function of the doping is generated from the $R_S(T)$ data. For each isothermal curve (a total of 30 different temperatures were used) a scaling factor $t$ is optimized such that it collapses onto $f$, chosen as the 12 K sheet resistance variation, as discussed below and shown by the dashed line. If the scaling is valid the corresponding experimental points of Fig. 4a should collapse onto the scaling function, as indeed observed in Fig. 4b.

In Supplementary Fig. 2 we show the same analysis between 7 and 20 K and remark that for higher temperatures (>15 K) thermal fluctuations may overshadow the QPT while at lower temperatures (<9 K) defect related weak localization phenomena may alter it, justifying our choice of the 12 K curve for the scaling function $f$. The power law relation between $T$ and $t$ from 9 to 15 K is shown in the inset of Fig. 4b. A perfect linear relationship is seen in a double logarithmic plot since all scaling factors $t$ should be of the form $t = T^{-1/\nu z}$. From the slope we obtain the critical exponent product $\nu z = 1.57 \pm 0.10$ similar to the $\nu z = 1.5$ found

in LSCO[7]. However in YBCO ($\nu z = 2.2$[8]) and BSCCO ($\nu z = 1.53$, 2.45, and 2.35 for three different devices[11]) contrasting results have been found.

## Discussion

The SIT has been uniquely characterized as having a non-zero and finite resistance phase corresponding to a "horizontal", temperature independent resistance[28] which is also the critical resistance. What is the nature of the SIT found here? The superconducting state is characterized by a complex order parameter. A continuous QPT is characterized by the continuous change of this order parameter across the transition (giving rise to power laws and scaling) and critical fluctuations of the amplitude or the phase of the order parameter at the transition. Amplitude fluctuations imply the breaking down of pairing above the critical temperature and the transformation of Cooper pair bosons to fermions as in superconductors described by the Bardeen–Cooper–Schieffer (BCS) scenario. The fermion state may be a metal or even an insulator if in the latter eventuality the fermions are localized by disorder or interactions.

Phase fluctuations in superconductors are typically described in the Berezinskii–Kosterlitz–Thouless scenario by the duality between Cooper pairs and vortices which are both bosons. In the superconducting state, vortices (associated with phase "slips" and dissipation), are bound in pairs and localized while Cooper pairs are mobile. The opposite situation prevails above the critical temperature and the system is insulating. If this duality is perfect, a simple argument[17] establishes the critical resistance threshold between the superconducting and insulating states at the quantum resistance with pair charge: $R_Q = h/(2e)^2 = 6.45 \, k\Omega. \square^{-1}$. However deviations from perfect duality generated by the nature of the interaction or factors like disorder are to be expected[28]. Thus the measurement of the critical resistance and the finite-scaling exponents of the continuous QPT can inform us about the all-important nature of the superconducting state. Simple models exist for some cases and are used for defining the universality class, for example $\nu z = 4/3$ in the classical percolation model, 7/3 in the quantum percolation model[13,29], and 2/3 in the 3D XY model[30]. Our result of $R_c = 6.85 \, k\Omega \pm 0.10 \, k\Omega$), which is reasonably close to $R_Q$ found in LSCO[7] favors the picture of a phase fluctuation driven transition and a strong coupling pairing interaction for high $T_c$ superconductivity at the underdoped limit as opposed to the weak coupling scenario in BCS superconductors. We find $\nu z = 1.57 \pm 0.10$ for the finite-scaling exponent product describing the measured transition. This again compares favorably with the value of 1.5[7] in LSCO and 1.53 in one BSCCO device[11], implying the same universality class for these compounds. A quantum critical point should imply that data at the lowest temperatures is the most indicative of the physics in question, so scaling of data at higher temperatures should be treated with caution. Just as quantum critical aspects may be washed out by thermal fluctuations, the ground state may also be overshadowed at lower temperatures by effects such as weak localization. Other effects such as a low temperature non-zero resistance metallic state instead of a superconducting one[31] or a double critical point[6] may also appear. However the clear separation in our data between the insulating and the superconducting regimes at the critical conductivity corresponding to $R_Q = h/(2e)^2$ provides support for the existence of the QPT and critical point[32].

We retain several other positives from this work. The use of a 1 u.c. sample ensures strict two dimensionality and makes a direct link with our earlier work on similar samples[9,10], notably with respect to the phase diagram and rigorous estimation of doping. The space charge doping method avoids problems which may arise in liquid dielectrics[16,33,34], ensures all measurements on a single, low disorder, good quality sample, and removes the sample dependent uncertainty that comes from chemical doping.

## Methods

**Sample preparation**. BSCCO precursors were exfoliated from bulk crystals, and deposited on soda-lime glass with a thickness of 0.5 mm. Then anodic bonding[23,35] was used to fabricate thin film BSCCO samples. The precursor on the glass substrate is placed between two electrodes and heated to ~180 °C to activate the Na$^+$ mobility. On the application of a negative gate voltage (~500 V) at the back side of the glass substrate, the Na$^+$ ions in the glass move away from the glass-sample interface, forming an O$^{2-}$ space charge at the sample-glass interface. This space charge sticks the first few nm of the precursor electrostatically on the glass substrate. Adhesive tape is used to exfoliate the precursor. A large area ultrathin BSCCO sample, the thickness of which is evaluated by Atomic Force Microscopy and optical contrast, is left on the glass surface. The sample is then annealed in air at 350 °C for 1 min to reduce its doping level by oxygen loss.

**Measurements**. Seventy nanometer thick gold contacts were evaporated onto the sample through a steel stencil mask to achieve a van der Pauw geometry device after the annealing process. We have found that both lithography and Cr buffer layers degrade sample quality. Sample quality and the absence of contamination are checked through the $R_S$ measurement. The glass substrate and device was then glued onto the backgate by silver paste (Fig. 1a). The four point sheet resistance measured at room temperature increased to ~7 k$\Omega$. $\square^{-1}$ compared to ~2 k$\Omega$. $\square^{-1}$ measured in similar devices prepared without annealing. The doping level was tuned inside a high-vacuum cryostat by space charge doping[23]. Above room temperature (350–380 K), Na$^+$ ion mobility inside the glass is activated. By applying a positive (or negative) gate voltage at the back of the glass substrate, the mobile Na$^+$ ions drift towards (or away from) the sample-substrate interface, creating a positive (or negative) space charge and corresponding electron (or hole) doping within the sample (Fig. 1a). The doping time at the doping temperature for moving between two doping levels varied from 10–110 min depending on the initial and final doping levels. This space charge is frozen on cooling down to room temperature or below loss of Na$^+$ ion mobility. In some earlier experiments[36–38] it has been shown that the effects of liquid ion gating induce oxygen drift in samples and a resulting chemical doping, perhaps more so than an electrostatic effect. In 1 u.c., strongly underdoped BSCCO this is not probable because of the absence of weakly bound or interstitial oxygen and a physical buffer region for oxygen drift. Moreover the space charge doping method works equally well in materials without any oxygen content[23,24].

The device was finally put into a high-vacuum Oxford He-flow cryostat for transport measurements between 6 and 350 K and Hall measurements. The measurements were performed with a DC current of 1 to 10 $\mu$A. An external magnetic field of up to 2 T perpendicular to the sample plane was supplied by a resistive electromagnet.

## Data availability
The relevant datasets are available on request from the corresponding authors

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

## Acknowledgements

We thank the Labex Matisse and the China Scholarship Council for support and Institut des NanoSciences de Paris for access to the electromagnet facility. We acknowledge the Consortium des salles blanches d'Ile de France, M. Rosticher and J. Palomo for access to clean room facilities.

## Author contributions

J.B. and A.S. designed the project. A.E. synthesized the BSCCO single crystal precursors, F.W. fabricated the devices and performed the measurements. F.W., J.B., and A.S. analyzed the data and wrote the paper.

## Competing interests

The authors declare no competing interests.
