## [Peer Review File · Nature Communications]

REVIEWER COMMENTS

Reviewer #1 (Remarks to the Author):

In this manuscript, F. Wang, et al. present an experimental study of the superconductor to insulator phase transition in ultrathin film layers of $\text{Bi}_{2.1}\text{Sr}_{1.9}\text{CaCu}_2\text{O}_{8+X}$ (BSCCO). Using a space charge doping technique they tune the electric properties of the material from the superconducting to the insulating regimes. They scale the resistance versus temperature (RT) data to obtain the product “ νz ” that is understood as the product of the correlation length exponent and the dynamic scaling exponent of the two dimensional quantum transition. Based on Hall measurements and the before mentioned RT experiments, they determine the doping concentration of the cuprate for the different applied gate voltages on their devices.

I don't suggest publishing this manuscript in Nature Communications. The set of results presented in this study are more suitable for a specialized journal of the field. The same authors have already published in Nature Communications the space charge doping technique for the same family of compounds and the results presented in this manuscript are complementary to many other studies published during the last decade. I miss as well an appropriate discussion about the behavior of the extra oxygen content of the cuprate in the space charged region of the cuprate flake. The authors use a pure electrostatic model that has been shown before to not fully describe the effects of the gate voltage on the samples under observation.

Reviewer #2 (Remarks to the Author):

Noteworthy results

A superconductor-insulator transition of the Bi 2212 high temperature superconductor was transitioned using the space charge doping technique.

Finite size scaling was used to analyze critical behavior and through the limiting resistance and the exponent product obtained. The limiting resistance, 6.85 ± 0.10 k Ω , that was found is close to

values obtained for LSCO Ref. 7), and the exponent product, 1.57 ± 0.10 is close to those found for BSCCO and LSCO (Refs. 11 and 7)

Significance.

The space-charge doping technique is a novel approach to modifying the carrier concentration in a thin film. The technique is nominally free of chemical effects that often occur using ionic liquids, and allows for substantially greater charge transfer than is possible using conventional dielectrics. This approach may serve in the investigation of other materials either ones that can be exfoliated or grown as thin films on glass substrates.

Does the work support the claims?

The major physics claim is that the scaling analysis supports the identification of the universality class of the quantum phase transition. However, all of these measurements, including those reported in the present manuscript were carried out at relatively high temperatures', between 9 K and 15K in the main manuscript and between 7 K and 20K in the Supplemental. The scaling, which is not terribly convincing, breaks down at the highest and lowest temperatures. The fact that it fails at the highest temperatures should be no surprise, as thermal fluctuations will drown out the quantum fluctuations associated with the quantum critical point which is at $T = 0$ K. The low temperature failure is another matter, and the data in Fig 1(b) suggests that something else is happening at low temperature. Reference 6 provides an example of what can happen. In this case, a second crossing point of the isotherms. It is also possible that some of the curves which appear to become superconducting at high temperatures flatten out as the temperature is reduced. Thus, the limiting low temperature behavior can be metallic for many values of the control parameter. All at temperatures below those reported in this work. (see: Kapitulnik et al., Rev. Mod. Phys. 91, 011002, (2019) in which "failed" superconductors are discussed. The bottom line is that a quantum phase transition occurs at zero temperature and scaling at relatively high temperatures must be treated carefully. A caveat weakening the conclusions would cover this problem.

Methodology

The scaling analysis is not up to the detailed treatment given by other works of a similar nature. However, the unique nature of the tuning method as applied to this problem is a strength.

The method of fabricating and characterizing the single layer samples needs to be clarified. As I understand it, first an exfoliated flake from bulk single crystal Bi 212 superconductor is anodically bonded to a glass substrate, and then the outer layers of. The flake are removed so that the sample is one unit cell of $\frac{1}{2}$ a unit cell in thickness. It is not clear as to how the thickness is determined, or whether the remaining unit cell has been contaminated in the anodic bonding process. This needs to be clarified

There is a substantial discussion of the method used to determine the hole concentration. How is what was ultimately used different from what other investigators have employed in the past in similar charging experiments'?

The authors should indicate the time needed at elevated temperature, to change the carrier concentration and thus to generate the experimental data.

Recommendation

The paper is well written, and it cites a suitable number of references to previous work. The study of the superconductor-insulator transition has entered a mature phase, as the subject is more than thirty years old. It has been revived in recent years by the development of techniques that allow for epitaxial growth and the exfoliation of very thin layers of layered compounds as well. As the use of various schemes for electrostatic doping. This work combines exfoliation with doping, and is deserving of publication. However, before publication, the issues discussed above need to be addressed.

Reviewer #3 (Remarks to the Author):

I find this work rather interesting. Using electro static doping, the authors were able to create a rather nice superconductor-insulator transition, in a single layer cuprates sample. This transition was studied in many other thin-film superconductor before, and also in cuprates materials with chemical doping. As was pointed out in the paper, the technique used by the authors have many advantages such as uniform doping and tunability. The authors were able to determine the critical point reliably and also extract the critical exponents. The data collapse near the critical point looks convincing.

I would recommend publication of this article. But I would recommend the authors to compare their results not only with previous experiments in cuprates, but also compare with previous SIT experiments on other thin film superconductors samples as well.

REVIEWER COMMENTS

Reviewer #1 (Remarks to the Author):

In this manuscript, F. Wang, et al. present an experimental study of the superconductor to insulator phase transition in ultrathin film layers of $\text{Bi}_{2.1}\text{Sr}_{1.9}\text{CaCu}_2\text{O}_{8+X}$ (BSCCO). Using a space charge doping technique they tune the electric properties of the material from the superconducting to the insulating regimes. They scale the resistance versus temperature (RT) data to obtain the product “ νz ” that is understood as the product of the correlation length exponent and the dynamic scaling exponent of the two dimensional quantum transition. Based on Hall measurements and the before mentioned RT experiments, they determine the doping concentration of the cuprate for the different applied gate voltages on their devices.

We thank the reviewer for accepting to review our manuscript and for their comments. Below we reply to them in detail.

I don't suggest publishing this manuscript in Nature Communications. The set of results presented in this study are more suitable for a specialized journal of the field. The same authors have already published in Nature Communications the space charge doping technique for the same family of compounds and the results presented in this manuscript are complementary to many other studies published during the last decade.

While the reviewer has expressed a negative opinion as to the suitability of the manuscript for this journal we argue below our case against the objections furnished. We have in fact published two papers previously in this journal where the same doping technique has been used, once before in 2017 on the same material. However none of these publications have been about the technique itself but about the physics of the concerned materials. Here we discuss the superconductor-insulator transition (SIT) in one unit cell BSCCO, which was not the subject of our earlier paper where we established the BSCCO phase diagram at a doping and temperature range much higher than the regime discussed here. The superconductor-insulator transition in two dimensions is a well-studied and important domain in its own right, as the reviewer states ('many other studies published during the last decade'), but those about high T_c superconductors are scarce. We have added a section in the manuscript about such studies in other compounds, as seen in the reply to reviewer 3. Our study stands out for the reasons we have stated in the manuscript and which are spelled out below:

- In high T_c superconductors, the universality class of the SIT is not established, nor is the fundamental question of the nature of the transition (phase coherence of already formed pairs or formation of pairs at the SIT) which has profound implications for the nature of high T_c superconductivity. Our results establish a close link with earlier results in LSCO and partial links with YBCO thus clearing the picture for the three principal families of high T_c superconductors.

- Electrostatic doping is an excellent technique for avoiding problems related to defects and sample multiplicity and inhomogeneity. We use our space charge doping technique for the first time in this context, all earlier studies have used liquid ion gating for which concerns exist including the one stated by the reviewer and addressed below. Our technique avoids several problems of the liquid ion technique and should be a pioneering experiment for several other problems as reviewer #2 has pointed out.

- Our results in 1 unit cell exfoliated BSCCO are in a perfectly 2D sample, which is important for the transition but also for the doping technique and its validity as discussed below. Earlier results have generally not been acquired in such well-defined 2D conditions.

- All doping experiments in high T_c materials are confronted with a central problem related to the quantitative estimation of the doping. In our work we extend a widely used method for estimating doping as a function of the critical temperature to the insulating region and show that the Hall Effect approach is not adequate. This was known and acknowledged by all earlier authors, but used nevertheless with sometimes erroneous results for want of a better method which we now provide. Notably in BSCCO the results using a Hall estimation of doping are not satisfactory, as we have already pointed out in our manuscript.

I miss as well an appropriate discussion about the behavior of the extra oxygen content of the cuprate in the space charged region of the cuprate flake. The authors use a pure electrostatic model that has been shown before to not fully describe the effects of the gate voltage on the samples under observation.

We assume that the reviewer is referring to studies in YBCO (Perez-Munoz et al. PNAS, 114, 215, 2017), SrTiO₃ (Li et al. Nano Lett 13, 4675, 2013) and VO₂ (Jeong et al. Science 339, 1402, 2013) which infer that in these experiments the effects of liquid ion gating induce oxygen drift in the samples and a resulting chemical doping. These articles do conclude however that the phase transitions observed are due to a real change in the doping. We show below that oxygen drift is not relevant to our case.

- In the material of the study most relevant to our sample (YBCO, Perez-Munoz et al.), it is well known that two structural entities exist in the YBCO unit cell, planes and chains. The oxygen in the chains is weakly bound and this is the oxygen which is accumulated or depleted while doping with stoichiometric change of oxygen. In the above reference it is explicitly stated that the observed oxygen diffusion related to strong electric fields is exclusively that of weakly bound chain oxygen atoms and not that of strongly bound oxygen atoms in the CuO₂ planes. In BSCCO no chains exist. Eventual weakly bound oxygen can be accumulated in interstitial oxygen in BiO planes, causing distortions and superstructure in the unit cell. There is quantitative information about this. Diffusion constants for oxygen have been measured in BSCCO (Runde et al. PRB, 45, 7375, 1992). In the a-b plane (movement of interstitial oxygen) they are five orders of magnitude higher than in the c direction, the direction of the space charge doping electric field. Moreover, our sample has been pre-annealed so that oxygen content is extremely low and corresponds to an initial chemical doping around $p=0.06$ leaving no possibility of weakly bound oxygen which may be moved by an electric field.
- In a liquid ion gated sample, the 2D sample is sandwiched between a substrate and the liquid ion layer. In our experiment, the sample is on the glass substrate which also serves as the gating material. Let us imagine that we induce oxygen diffusion in the sample. Even if weakly bound oxygen is available, it cannot diffuse in and out of a 1 unit cell sample into the vacuum or the glass, reversibly. In a 3 unit cell sample sandwiched sample (Perez-Munoz et al.) oxygen diffusion can be imagined between different layers of the thicker sample or eventually at or with the sample/liquid ion interface. In our case this is not possible.
- A further guaranty comes from the fact that we have successfully doped, with precisely the same method and similar doping changes, samples such as graphene [21] and MoS₂ [24] which do not contain oxygen.

We have added the following text to the manuscript:

In some earlier experiments [34-36] it has been shown that the effects of liquid ion gating induce

oxygen drift in samples and a resulting chemical doping, perhaps more so than an electrostatic effect. In one unit cell, strongly under-doped BSCCO this is not probable because of the absence of weakly bound or interstitial oxygen and a physical buffer region for oxygen drift. Moreover the space charge doping method works equally well in materials without any oxygen content [21,24]

Reviewer #2 (Remarks to the Author):

Noteworthy results

A superconductor-insulator transition of the Bi 2212 high temperature superconductor was transited using the space charge doping technique.

Finite size scaling was used to analyze critical behavior and through the limiting resistance and the exponent product obtained. The limiting resistance, 6.85 ± 0.10 kOhm, that was found is close to values obtained for LSCO Ref. 7), and the exponent product, 1.57 ± 0.10 is close to those found for BSCCO and LSCO (Refs. 11 and 7)

Significance.

The space-charge doping technique is a novel approach to modifying the carrier concentration in a thin film. The technique is nominally free of chemical effects that often occur using ionic liquids, and allows for substantially greater charge transfer than is possible using conventional dielectrics. This approach may serve in the investigation of other materials either ones that can be exfoliated or grown as thin films on glass substrates.

We thank the reviewer for accepting to review our manuscript, their appreciation of our work and the constructive criticism. We appreciate the recognition of novelty, potential and the recommendation to publish.

Does the work support the claims?

The major physics claim is that the scaling analysis supports the identification of the universality class of the quantum phase transition. However, all of these measurements, including those reported in the present manuscript were carried out at relatively high temperatures', between 9 K and 15K in the main manuscript and between 7 K and 20K in the Supplemental. The scaling, which is not terribly convincing, breaks down at the highest and lowest temperatures. The fact that it fails at the highest temperatures should be no surprise, as thermal fluctuations will drown out the quantum fluctuations associated with the quantum critical point which is a $t_c = 0$ K. The low temperature failure is another matter, and the data in Fig 1(b) suggests that something else is happening at low temperature. Reference 6 provides an example of what can happen. In this case, a second crossing point of the isotherms. It is also possible that some of the curves which appear to become superconducting at high temperatures flatten out as the temperature is reduced. Thus, the limiting low temperature behavior can be metallic for many values of the control parameter. All at temperatures below those reported in this work. (see: Kapitulnik et al., Rev. Mod. Phys. 91, 011002, (2019) in which "failed "superconductors are discussed. The bottom line is that a quantum phase transition occurs at zero temperature and scaling at relatively high temperatures must be treated carefully. A caveat weakening the conclusions would cover this problem.

The validity of the scaling argument and possible non-trivial behaviour at lower than measured temperatures are important points and we thank the reviewer for bringing them up and suggesting directions for explaining them in a more detailed way. These are also complex points involving several possible phenomena and explanations many of which are discussed in the reference provided by the

reviewer which we have cited in the revised version. Below we discuss these points and justify our choices in the light of these points:

a) The zero temperature quantum critical point should, as remarked by the reviewer, imply that data at the lowest temperatures is the most indicative of the physics in question. However just as quantum critical aspects may be blurred on the high temperature side by thermal fluctuations, the ground state may also be overshadowed on the low temperature side by effects such as weak localisation which can become dominant before this ground state can be reached through the critical point and across the quantum phase transition. This or other analogous effects will always be present in a real sample. So we argue for a domain of temperatures not too low and not too high where the effects of the quantum phase transition are the most evident. We also argue that it is the scaling process which allows us to determine this domain as shown by the two ranges we have used (7K-20K in the supplement and 9K- 15K in the manuscript). In particular for the data in the range 7K to 9K, it can be clearly seen (fig2 of supplementary material), that νz (which is given by the local slope) is much higher than 1.5. This directly results from the R_s of the insulating phases rising higher than the 'expected value' for temperatures below 9K and indicating that it is indeed a localisation like phenomenon which causes this discrepancy. In high T_c samples such behaviour is clearly visible in earlier work (for example Leng et al. reference 8)

b) We cannot exclude the possibility of a flattening of the superconducting dip at lower temperatures and a quantum superconductor-metal transition invoked by Kapitulnik et al. (and shown for example for LSCO data in fig 15 of their work). But even this transition and interpretation is only valid till a certain low temperature limit. There is no guaranty that R_s stays finite at still lower and unattained temperatures and in fact for 2D samples it should become infinite as temperature is continuously lowered. As the reviewer has pointed out and as shown in reference 6 of the earlier manuscript, other phenomena may also kick in as the temperature scale is changed. However, as successfully argued in reference 6 this could mean double critical points with no hierarchy of one critical point over the other but a change in criticality according to the coherence length picked out by the temperature and the control parameter in an *inhomogeneous* sample.

c) In high T_c samples with pairing in the underdoped region expected to persist to high temperatures and a high T_c at optimal doping, we can expect quantum critical effects to persist at temperatures higher than in metals where a superconducting transition is achieved at a few K or a few tenths of a K. Indeed in YBCO (Leng et al. reference 8) the low temperature data is not used for scaling either.

d) Again quoting from Kapitulnik et al.: "In [...] cases, where a clear separation between the normal state and the regime of superconducting fluctuations is observed, the notion that the critical conductivity is associated with a self-dual point of charge $2e$ bosons, and hence has a value $= h/(2e)^2$, has some experimental support". This is our case.

To conclude we fully accept the word of caution of the reviewer that 'scaling at higher temperatures must be treated cautiously' and have included parts of the above discussion in the conclusion of the manuscript:

A quantum critical point should imply that data at the lowest temperatures is the most indicative of the physics in question, so scaling of data at higher temperatures should be treated with caution. Just as quantum critical aspects may be washed out by thermal fluctuations, the ground state may also be overshadowed at lower temperatures by effects such as weak localisation. Other effects such as a low temperature non-zero resistance metallic state instead of a superconducting one at lower temperatures [30] or a double critical point [6] may also appear. However the clear separation in our data between the insulating and the superconducting regimes at the critical conductivity

corresponding to $R_Q = h/(2e)^2$ provides support for the existence of the quantum phase transition and critical point [32].

Methodology

The scaling analysis is not up to the detailed treatment given by other works of a similar nature. However, the unique nature of the tuning method as applied to this problem is a strength.

We thank the reviewer for appreciating the tuning method and confirm that given some complications in the liquid ion gating method (possibility of intercalation, purity of the ionic liquid, mechanical stress on sample due to the freezing of the liquid) our method is a promising alternative. We would like to stress that our scaling analysis is rigorous and careful and equivalent to that in earlier published work. However we have redone the analysis on a much finer temperature grid by increasing the number of isothermal sheet resistance curves used in the scaling by a factor of five. Fortunately we find that this does not alter the critical exponents, nor the critical doping or sheet resistance that we found earlier. We give below the modified account of the analysis which has been added to the manuscript. Figure 4 has also been consequently modified.

In Fig. 4b we investigate the existence of a single finite-size scaling function f for data between 9K and 15K by using as the abscissa $|p-p_c|t$. Isothermal sheet resistance data as a function of the doping is generated from the $R_S(T)$ data. For each isothermal curve (a total of 30 different temperatures were used) a scaling factor t is optimized such that it collapses onto f , chosen as the 12K sheet resistance variation, as discussed below and shown by the dashed line. If the scaling is valid the corresponding experimental points of Fig. 4a should collapse onto the scaling function, as indeed observed in Fig.4b. In Supplementary Fig. 2 we show the same analysis between 7K and 20K and remark that for higher temperatures ($> 15K$) thermal fluctuations may overshadow the QPT while at lower temperatures ($< 9K$) defect related weak localization phenomena may alter it, justifying our choice of the 12K curve for the scaling function f . The power law relation between T and t from 9K to 15K is shown in the inset of Fig. 4b.

The method of fabricating and characterizing the single layer samples needs to be clarified. As I understand it, first an exfoliated flake from bulk single crystal Bi 212 superconductor is anodically bonded to a glass substrate, and then the outer layers of. The flake are removed so that the sample is one unit cell of $\frac{1}{2}$ a unit cell in thickness. It is not clear as to how the thickness is determined, or whether the remaining unit cell has been contaminated in the anodic bonding process. This needs to be clarified

The thickness of the sample is verified in a first step by checking the optical contrast and in a further step by an AFM measurement. The optical contrast estimation of thickness by a trained eye is found to be coherent with the AFM measurement in all our samples. The step from a 1 unit cell to a functioning device is more complicated. As explained in the methods section, we evaporate gold through a shadow mask and without any buffer layer (Cr layer for example) to contact the sample. We have found that both lithography and Cr buffer layers degrade sample quality. The best check of the sample quality and the absence of contamination is the R_S measurement, the optimal T_c and the width of the transition.

We have modified the text related to sample fabrication in the methods section.

A large area ultra-thin BSCCO sample, the thickness of which is evaluated by Atomic Force Microscopy and optical contrast, is left on the glass surface. The sample is then annealed in air at 350°C for 1 minute to reduce its doping level by oxygen loss.

Measurements. 70 nm thick gold contacts were evaporated onto the sample through a steel stencil mask to achieve a van der Pauw geometry device after the annealing process. We have found that both lithography and Cr buffer layers degrade sample quality. Sample quality and the absence of contamination are checked through the R_S measurement.

There is a substantial discussion of the method used to determine the hole concentration. How is what was ultimately used different from what other investigators have employed in the past in similar charging experiments'?

Other investigators directly used the Drude formula to estimate the hole concentration as equal to $S/R_s(T_s)$ where T_s is some temperature in the normal state (100K or 200K) and S a fitted constant. While this may work approximately in some cases, it is illusory to expect that the carrier concentration in high T_c materials can be estimated with such a simple approach and in the case of BSCCO it duly fails. We use an empirical estimation of the carrier concentration, based on the variation of T_c with doping (the dome shape) and extend it to the non-superconducting regime for the first time, as explained in the manuscript. We obtain carrier concentrations coherent with the generic phase diagram.

The authors should indicate the time needed at elevated temperature, to change the carrier concentration and thus to generate the experimental data.

The doping time at the doping temperature for moving between two doping levels varied from 10- 110 minutes depending on the initial and final doping levels. This has been added to the methods section.

Recommendation

The paper is well written, and it cites a suitable number of references to previous work. The study of the superconductor-insulator transition has entered a mature phase, as the subject is more than thirty years old. It has been revived in recent years by the development of techniques that allow for epitaxial growth and the exfoliation of very thin layers of layered compounds as well. As the use of various schemes for electrostatic doping. This work combines exfoliation with doping, and is deserving of publication. However, before publication, the issues discussed above need to be addressed.

We would again like to thank the reviewer for highlighting in a nutshell the importance and the background to our work. We have replied to all questions above.

Reviewer #3 (Remarks to the Author):

I find this work rather interesting. Using electro static doping, the authors were able to create a rather nice superconductor-insulator transition, in a single layer cuprates sample. This transition was studied in many other thin-film superconductor before, and also in cuprates materials with chemical doping. As was pointed out in the paper, the technique used by the authors have many advantages such as uniform doping and tunability. The authors were able to determine the critical point reliably and also extract the critical exponents. The data collapse near the critical point looks convincing.

I would recommend publication of this article. But I would recommend the authors to compare their results not only with previous experiments in cuprates, but also compare with previous SIT experiments on other thin film superconductors samples as well.

We thank the reviewer for accepting to review our manuscript, their appreciation of our work and the suggestions. We appreciate the recognition of novelty, potential and the recommendation to publish. We have added the requested additional comparisons below, including new references.

Superconductor-insulator transitions have been induced in a variety 2D superconductors by tuning different external parameters and studied with a finite-size scaling analysis. There is however not much uniformity in the findings as both the superconducting systems and the tuning parameters are diverse. Magnetic field is a common, easily tunable and accessible external parameter. Magnetic field driven SIT studies have reported a large variety of critical exponents and resistance at criticality: from $\nu z \sim 0.67$ and $R_c = 1.3 \text{ k}\Omega$ in NbSi thin films [12] to $\nu z \sim 2.3$ and $R_c = 6 \text{ k}\Omega$ in Indium Oxide thin films [13]. Furthermore, in quenched condensed bismuth thin films the magnetic field driven SIT was shown to have $\nu z \sim 0.7$ and $R_c = 8 \text{ k}\Omega$, while in the same system the thickness driven SIT displayed $\nu z \sim 1.2$ around the same critical resistance, indicating a fundamental difference in their nature [3]. It is therefore difficult to come to a general conclusion about the SIT in different systems and induced by different tuning parameters. The high critical magnetic fields of most high critical temperature superconductors renders the study of the magnetic field driven SIT difficult (with the exception of the electron doped cuprate NdCeCuO [14]). However, improvements in extreme electrostatic doping techniques have provided access to the carrier density driven SIT in LSCO and YBCO.

REVIEWERS' COMMENTS

Reviewer #1 (Remarks to the Author):

I appreciate the efforts of the authors to reply to my criticism.

The reviewed version of the paper describes the physics problem with a broader point of view that makes the paper more interesting for a broader community.

My technical discussion about the oxygen displacements in the sample has been properly discussed in the reviewed version of the paper as well.

Based on the above, I recommend the publication of the reviewed version of the article.

Reviewer #2 (Remarks to the Author):

The authors have addressed the concerns of all the reviewers in a satisfactory and professional manner. Although I don't agree with every one of their arguments, the claims that are made are reasonable. I recommend that the references be checked to be certain that they all have page numbers, and that this work be published.